# Quiet-STaR: Language Models Can Teach Themselves to Think Before Speaking

**Eric Zelikman**
Stanford University
ezelikman@cs.stanford.edu

**Georges Harik**
Notbad AI Inc
georges@gmail.edu

**Yijia Shao**
Stanford University
shaoyj@cs.stanford.edu

**Varuna Jayasiri**
Notbad AI Inc
vpjayasiri@gmail.com

**Nick Haber**
Stanford University
nhaber@stanford.edu

**Noah D. Goodman**
Stanford University
ngoodman@stanford.edu

## Abstract

When writing and talking, people sometimes pause to think. Although reasoning-focused works have often framed reasoning as a method of answering questions or completing agentic tasks, reasoning is implicit in almost all written text. For example, this applies to the steps not stated between the lines of a proof or to the theory of mind underlying a conversation. In the Self-Taught Reasoner (STaR, Zelikman et al. 2022), useful thinking is learned by inferring rationales from few-shot examples in question-answering and learning from those that lead to a correct answer. This is a highly constrained setting – ideally, a language model could instead learn to infer unstated rationales in arbitrary text. We present **Quiet-STaR**, a generalization of STaR in which LMs learn to generate rationales at each token to explain future text, improving their predictions. We address key challenges, including 1) the computational cost of generating continuations, 2) the fact that the LM does not initially know how to generate or use internal thoughts, and 3) the need to predict beyond individual next tokens. To resolve these, we propose a tokenwise parallel sampling algorithm, using learnable tokens indicating a thought's start and end, and an extended teacher-forcing technique. Encouragingly, generated rationales disproportionately help model difficult-to-predict tokens and improve the LM's ability to directly answer difficult questions. In particular, after continued pretraining of an LM on a corpus of internet text with Quiet-STaR, we find zero-shot improvements on GSM8K (5.9%→10.9%) and CommonsenseQA (36.3%→47.2%) and observe a perplexity improvement of difficult tokens in natural text. Crucially, these improvements require no fine-tuning on these tasks. Quiet-STaR marks a step towards LMs that can learn to reason in a more general and scalable way.

> "Life can only be understood backwards; but it must be lived forwards."
>
> — Søren Kierkegaard

## 1 Introduction

Much of the meaning of text is hidden between the lines: without understanding why statements appear in a document, a reader has only a shallow understanding. Moreover, this has been repeatedly shown to be true for LMs as well, in the contexts of tasks ranging from commonsense reasoning to theorem proving to programming (Wei et al., 2022b; Nye et al., 2021; Zelikman et al., 2022; 2023a; Kojima et al., 2022). Reasoning about implications of text to predict later text has consistently been shown to improve LM performance on a variety of tasks, but methods for allowing LMs to learn from their reasoning (e.g., Zelikman et al. 2022) have focused on solving individual tasks or predefined sets of tasks (e.g., Wei et al. 2021b). These works rely on carefully curated datasets to provide either specific reasoning tasks or in some cases, the reasoning itself. We instead ask, if reasoning is implicit in all text, why shouldn't we leverage the task of language modeling to teach reasoning?

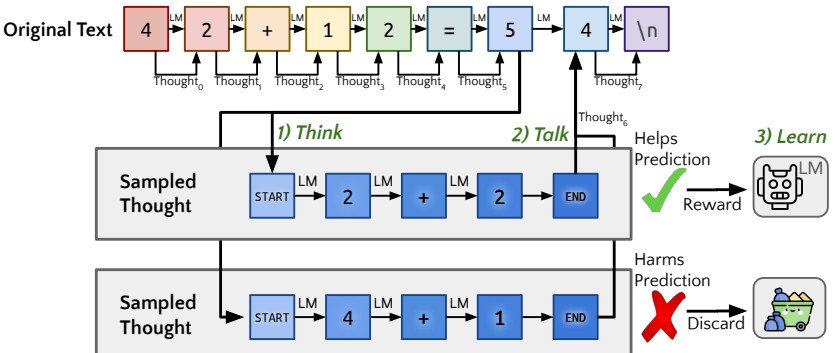

Figure 1: **Quiet-STaR**. We visualize the algorithm as applied during training to a single thought. We generate thoughts, in parallel, following all tokens in the text (think). The model produces a mixture of its next-token predictions with and without a thought (talk). We apply REINFORCE, as in STaR, to increase the likelihood of thoughts that help the model predict future text while discarding thoughts that make the future text less likely (learn). Note the second sampled thought (4 + 1) represents one that is *not* useful in predicting "4".

In particular, the Self-Taught Reasoner (STaR, Zelikman et al. 2022) showed that LMs can bootstrap their reasoning ability on question-answering (QA) datasets by sampling rationales to attempt to answer questions, training on rationales if they led to a correct final answer, and then repeating this to iteratively solve more difficult problems. Yet, training from curated QA datasets limits the scale and generalizability of the rationales. QA datasets, especially high-quality ones, require thoughtful curation and will inherently only ever cover a subset of reasoning tasks. Thus, we extend STaR – instead of the LM learning to reason on particular tasks like mathematical QA, we train an LM to generate reasoning that helps it infer future text from a large internet text corpus. As a result, we allow the LM to learn from the diverse tasks present in language (Weber et al., 2021). This builds on an intuition essential to the current language modeling paradigm, namely, that "language models are unsupervised multitask learners" (Radford et al., 2019). Thus, as in STaR, we leverage the LM's pre-existing reasoning ability to generate rationales and train the LM on them with a REINFORCE-based reward (Williams, 1992). We refer to this technique as Quiet-STaR, as it can be understood as applying STaR "quietly", training the model to think before it speaks.

Broadly, Quiet-STaR proceeds by generating rationales after every token to explain future text (*think*), mixing the future-text predictions with and without rationales (*talk*), and then learning to generate better rationales using REINFORCE (*learn*). We apply Quiet-STaR to Mistral 7B (Jiang et al., 2023) using the web text datasets OpenWebMath (Paster et al., 2023) and Colossal Clean Crawled Corpus (C4, Raffel et al. 2020). We find that, even without dataset-specific fine-tuning, Quiet-STaR results in improvements to zero-shot direct-reasoning abilities on CommonsenseQA (36.3%→47.2%) and GSM8K (5.9%→10.9%), and that these improvements consistently increase with the number of tokens used in the LM's internal thoughts. Lastly, we qualitatively investigate patterns in the generated rationales.

In solving this task, we make the following contributions:

1. We generalize STaR to learn reasoning from diverse unstructured text data. To our knowledge, this is the first work explicitly training LMs to **reason generally** from text, rather than on curated reasoning tasks or collections of reasoning tasks.
2. We propose and implement a **parallel sampling algorithm** that makes our training procedure scalable, generating rationales from all token positions in a given string.
3. We introduce custom **meta-tokens** at the start and end of each thought to allow the LM to learn that it should be generating a rationale and when it should make a prediction based on that rationale.
4. We apply a **mixing head** to retrospectively determine how much to incorporate the next-token prediction from a given thought into the current next-token prediction.
5. We show that a **non-myopic loss**, including multiple tokens ahead for language modeling, improves the effect of thinking.
6. On multiple tasks, we demonstrate that thinking allows the LM to predict difficult tokens better than one trained on the same web text, improving with longer thoughts.

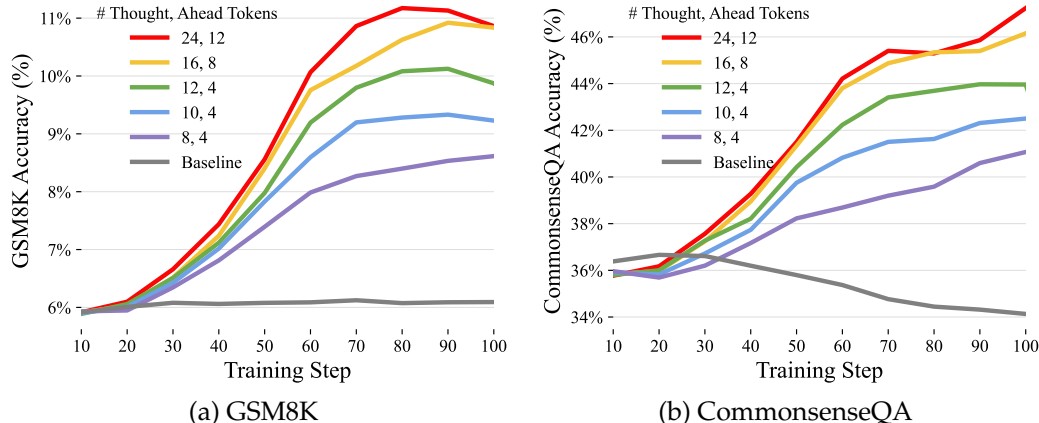

Figure 2: **Generalization Results**. We evaluate the extent to which the model trained with Quiet-STaR generalizes to directly answering problems that require reasoning. The left plot (a) shows the zero-shot accuracy on GSM8K, while the right plot (b) shows the zero-shot accuracy on CommonsenseQA, without any fine-tuning. In both plots, the x-axis represents training steps, and each line corresponds to a different number of thinking tokens used during Quiet-STaR training. The y-axis measures the zero-shot direct accuracy on the respective datasets. We also include an inference normalized version of this plot in Figure 6. Note the baseline corresponds to naive continued pretraining.

## 2 Related Work

### 2.1 Reasoning in Language Models

There have been many works on training and exploiting language models to solve difficult tasks by first training them to reason through them. For example, Rajani et al. (2019) fine-tuned a language model on human-written reasoning traces placed before multiple-choice answers to commonsense reasoning questions, finding it outperformed a language model trained to directly predict the answers. Shwartz et al. (2020) demonstrated that language models, when provided with some scaffolding, can generate these helpful chain-of-thought solutions without additional supervision. Later, Nye et al. (2021) demonstrated that "scratchpads" required less scaffolding when the language models were more capable, a result later reinforced by Wei et al. (2022b), emphasizing informal tasks, and further strengthened by Kojima et al. (2022), demonstrating this behavior could be accomplished zero-shot. Most recently, Wang & Zhou (2024) showed further that for commonsense-question answering, one could force a language model to leverage chain-of-thought reasoning by preventing it from emitting any valid answer tokens unless it was confident. However, once again, these approaches only work for a question-answer dataset, and Wang & Zhou (2024) relies on heuristics to identify when the model has output answer tokens. Somewhat like TRICE (Phan et al., 2023), we use the relative improvements in the log-likelihood of the target text across rationales as an estimate of quality, but we simply subtract the mean reward and do not incorporate more complex control variates.

### 2.2 Training Language Models to Reason

One direction that researchers have used to train language models to reason or improve their reasoning is training the language model on mined reasoning traces or reasoning-like data (Rajani et al., 2019; Wei et al., 2021a; Lewkowycz et al., 2022; Chung et al., 2022; Gunasekar et al., 2023). Although this approach has been demonstrated to be effective, it comes with drawbacks. It requires either manual annotation, which is sensitive to the capability of the annotators and is off-policy for the language model (i.e., the distribution of reasoning is not text that the language model would otherwise likely have generated). This approach is also expensive, difficult to scale, and provides no clear path to solving problems harder than those that the annotators are capable of solving.

---

**Algorithm 1:** Quiet Self-Taught Reasoner (Quiet-STaR)

---

**Input:** Language model $\theta_0$, training steps num_steps, sequence length $l$, thought length $t$, learning rate $\alpha$, batch size $b$, number of thoughts $n_{thoughts}$, number of ground truth tokens used for supervising each thought $n_{true}$

**Output:** Language model $\theta$ that generates rationales to predict future text

**for** $i = 0$ **to** num_steps **do**

    Sample batch of sequences $X$ of length $l$

    $h^{init} \leftarrow$ hidden_states$_{\theta_i}(X)$

    **for** $j = 1$ **to** $l$ **in parallel using attention mask do**

        $\log p^{init}_{j:j+n_{true}} \leftarrow$ lm_head$_{\theta_i}(h^{init}_{j:j+n_{true}})$         `// Predict next tokens`

        $T_j \leftarrow$ generate_tokens$_{\theta_i}([X_{:j};$<start_thought>$], t, n_{thoughts})$ `// Generate thought`

        $T_j \leftarrow [T_j;$<end_thought>$]$

        $h^{thought}_{j:j+n_{true}} \leftarrow$ hidden_states$_{\theta_i}([X_{:j}; T_j; X_{j:j+n_{true}-1}])$

        $\log p^{thought}_{j:j+n_{true}} \leftarrow$ lm_head$_{\theta_i}(h^{thought}_{j:j+n_{true}})$       `// Predict next tokens w/ thought`

        $w_{j:j+n_{true}} \leftarrow$ mixing_head$_{\theta_i}(h^{thought}_{j:j+n_{true}}, h^{init}_{j:j+n_{true}})$

        $\log p^{talk}_{j:j+n_{true}} \leftarrow w_{j:j+n_{true}} \cdot \log p^{init}_{j:j+n_{true}} + (1 - w_{j:j+n_{true}}) \cdot \log p^{thought}_{j:j+n_{true}}$ `// Mix logits`

        $\mathcal{L}^{NLL}_j \leftarrow -\log p^{talk}_{j:j+n_{true}}(X_{j+1:j+n_{true}+1})$

        $r_j = \log p^{talk}_{j:j+n_{true}}(X_{j+1:j+n_{true}+1}) - \log \overline{p}^{talk}_{j:j+n_{true}}(X_{j+1:j+n_{true}+1})$

        $\nabla_\theta \mathcal{L}^{REINFORCE}_j \leftarrow -r_j \mathbb{1}[r_j > 0] \cdot \nabla_\theta \log p_{\theta_i}(T_j | [X_{:j};$<start_thought>$])$

        $\nabla_\theta \mathcal{L}_j \leftarrow \nabla_\theta \mathcal{L}^{NLL}_j + \nabla_\theta \mathcal{L}^{REINFORCE}_j$

    $\theta_{i+1} \leftarrow \theta_i - \alpha \sum_{j=1}^{l} \nabla_\theta \mathcal{L}_j$         `// Update model parameters`

**return** $\theta_{num\_steps}$

---

Another direction for teaching reasoning relies on a language model's own generated reasoning, which can be seen as building on a large body of literature on self-play (Silver et al., 2017; Anthony et al., 2017; Polu & Sutskever, 2020). These include methods such as the Self-Taught Reasoner (Zelikman et al., 2022), which demonstrated that a language model iteratively trained on its reasoning that led to correct answers could solve increasingly difficult problems. Later work aimed to leverage additional information or assumptions such as Huang et al. (2022) which demonstrated that the algorithm proposed in STaR could still work if one assumed that the majority-vote answer was correct (although this has a lower ultimate performance). Further work has generalized the results of Zelikman et al. (2022), such as Uesato et al. (2022) which demonstrated additional usefulness to "process-based" supervision where incorrect reasoning traces were filtered, recently V-STaR (Hosseini et al., 2024) that demonstrates that training a verifier to guide generation also improves performance, as well as TRICE (Hoffman et al., 2024) which maximizes the marginal likelihood of the correct answer given several reasoning traces per problem. Finally, related work has also explored learning intermediate reasoning in the constrained setting of making mathematical statements, where statements in the model's intermediate reasoning could be constrained to only be valid mathematical statements (Poesia et al., 2023). We include further discussion of related reasoning works in Appendix F.

### 2.3 Meta-tokens

Recently, a growing body of work has demonstrated the usefulness of custom tokens optimized to perform specific functions in the context of a neural network – for this reason, they have also been referred to as "function vectors." (Todd et al., 2023). One of the original instantiations of this was prompt-tuning (Lester et al., 2021) (and relatedly prefix-tuning (Li & Liang, 2021)), where the embeddings corresponding to the tokens of a prompt could be optimized to better accomplish a task. Others have applied meta-tokens to compress long prompts (Li et al., 2023; Jung & Kim, 2023) for efficiency. Most relevant to this work, Mu et al.

(2024) optimized a token such that, when the tokens after it could not attend to the tokens before it (i.e., a context compression token), it would provide sufficient information to future tokens. Although we do not focus on compression, we share the problem of learning a token that affects attention and controls complex downstream behavior. In one related work, Goyal et al. (2023) show that learning a single "pause" token (essentially representing each token as two tokens) improves LM performance. However, unlike the thought tokens in our work, this pause token does not initialize a thought – instead, it can be seen as acting as the entirety of the thought. We find that reasoning in language is significantly more helpful.

## 3 Problem Statement

In this work, we introduce an auxiliary 'rationale' variable between each pair of observed tokens of the sequence. We then aim to optimize a language model with parameters $\theta$ with the capacity to generate intermediate thoughts (or rationales) such that

$$\theta^* = \arg\max_\theta E_x \left[ log\, p_\theta \left( x_{i:n} | x_{0:i}, \text{rationale}_\theta \left( x_{0:i} \right) \right) \right]$$

Note that, in principle, this provides no advantage over an optimal language model that already correctly models the language's distribution over strings. Yet, in practice, extensive prior work has shown that language models benefit from intermediate rationales on reasoning tasks (Nye et al., 2021; Zelikman et al., 2022; Wei et al., 2022b). Some work has aimed to explain the effects of chain-of-thought reasoning, namely attributing it to "locality of experience" (Prystawski et al., 2024). More broadly, reasoning allows a model to decompose a challenging computation into smaller steps. In effect, we train the model to learn which decomposition and planning steps are effective in predicting future text. Also note that we formulate the objective as accurately predicting the remaining sequence, rather than only the next token. Once again, for an optimal LM these would be equivalent. However, we find that the non-myopic formulation leads to a more effective loss for learning rationales.

## 4 Quiet-STaR

### 4.1 Overview

Quiet-STaR operates with three main steps (Figure 1):

1. **Parallel rationale generation (think, Subsection 4.2)**: In parallel across $n$ tokens $x_i$ in an input sequence $x_{0:n}$, we generate $r$ rationales of length $t$: $c_i = (c_{i1}, \ldots, c_{it})$, resulting in $n \times r$ rationale candidates. We insert learned <|startofthought|> and <|endofthought|> tokens to mark each rationale's start and end.
2. **Mixing post-rationale and base predictions (talk, Subsection 4.3)**: From the hidden state output after each rationale, we train a "mixing head" – a shallow MLP producing a weight determining how much the post-rationale next-token predicted logits should be incorporated compared to the base language model predicted logits. This approach eases distribution shift early in finetuning, due to introducing rationales.
3. **Optimizing rationale generation (learn, Subsection 4.4)**: We optimize the rationale generation parameters (start/end tokens and LM weights) to increase the likelihood of rationales that make future text more probable. We use REINFORCE to provide a learning signal to rationales based on their impact on future-token prediction. To reduce variance, we apply a teacher-forcing trick to include in the loss the likelihood of predicting not only the token after the thought but also later tokens.

### 4.2 Parallel Generation

A key challenge in Quiet-STaR is efficiently generating rationales at each token position in the input sequence. Naively, this would require a separate forward pass for each token, which becomes computationally intractable for long sequences.

We allow for highly parallel generation by first observing that an inference pass of a language model produces a probability distribution over the next tokens for all input tokens. Naturally,

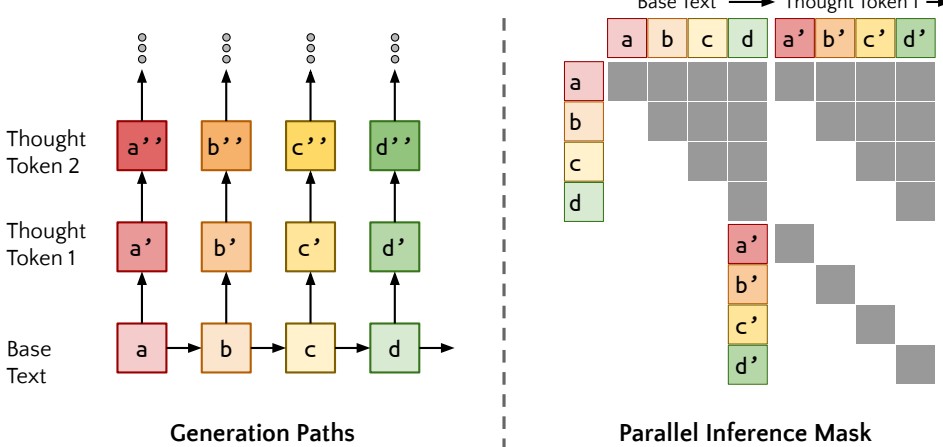

Figure 3: **Parallel Generation**. By constructing an attention mask that allows all thought tokens to pay attention to themselves, all preceding thought tokens within the same thought, and the preceding text, we can generate continuations of all of the thoughts in parallel. Each inference call is used to generate one additional thought token for all text tokens.

this allows us to sample one next token from each token in the input. If one has generated a successor from each token, it is not possible to simply continue with the original sequence. For example, imagine predicting the next token after each token of "$< bos >$ the cat sat" one might generate "yes orange saw down" – each successor by itself is a reasonable next token to a prefix of the sequence, but the list of tokens is a set of "counterfactual" continuations of these prefixes. We can, however, leverage these continuations to generate hidden thoughts for each observed token.

To do this efficiently, we cache each forward pass and concatenate a diagonal attention mask to the previous attention mask: each generated token now attends to all of the tokens that were used to generate it, as well as to itself (but not to token on other "counterfactual" paths). Moreover, this parallelized next-sampling token procedure can be repeated arbitrarily many times (or at least, until one runs out of memory). We visualize this procedure in Figure 3 and highlight additional ways to make this algorithm faster in Appendix B. In particular, in Figure 3 (left) each vertical bottom-to-top iteration corresponds to one horizontal (left-to-right) iteration of Figure 3 (right).

### 4.3 "Mixing" (Residual) Heads

When starting with a pre-trained model, thoughts will initially be out of distribution, and hence harm language modeling performance. To smooth the transition to thinking, we introduce a learned interpolation between the LM predictions with and without thoughts. Given the end-of-thought token's hidden state and the hidden state of the original text token, the mixing head outputs a weight that determines the extent to which the post-thought prediction logits will be used. We use a shallow multi-layer perceptron for this head, outputting a scalar for each token. We include implementation details in Appendix A.

### 4.4 Optimizing Rationale Generation

#### 4.4.1 Optimizing Start-of-Thought and End-of-Thought Tokens

The $<|startofthought|>$ and $<|endofthought|>$ tokens serve as learned meta-tokens that control the model's rationale generation. Optimizing the representation of these tokens, especially the $<|startofthought|>$ token, is crucial but challenging due to the discrete nature of the rationale tokens. We initialize the start and end token embeddings to the embedding corresponding to the em dash, "−−−", which often appears in text data to denote a pause or thought. This leverages the language model's preexisting knowledge. In addition, to allow these embeddings to be optimized more quickly, we apply a (hyperparameter) weight to

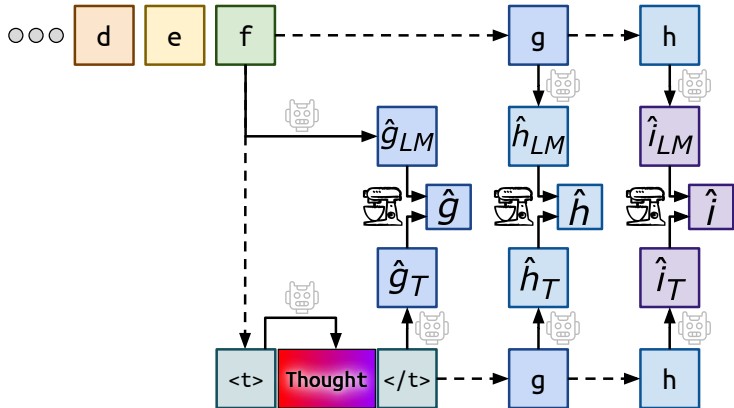

Figure 4: **Forward Pass and Teacher Forcing**. We visualize a single forward pass of our algorithm. Solid lines denote language model computation, while dashed lines indicate tokens are inserted via teacher forcing, and the mixer represents the mixing head. In particular, we visualize predicting three tokens ahead. Thought generation is shown in more detail in Figure 1 and Figure 3.

the gradients of these embeddings during the update step. Intuitively, the start thought tokens can be understood as putting the model into a "thinking mode" and the end thought token can be understood as telling the model when it's done thinking.

### 4.4.2  Non-myopic Scoring and Teacher-forcing

Because we do not expect thoughts to be useful in predicting every token, we would prefer the model's reward to depend less on the exact next word in the text following the thought and more on the following semantic content. There are two primary challenges here. First, unlike in typical language modeling with transformers, only the thoughts corresponding to a given next-token prediction receive a gradient from that prediction—a consequence of our parallel sampling strategy. We could address this by adding loss terms for future tokens by sampling the tokens before. However this would result in much higher entropy for language modeling in general and lower-quality generated text, because it would train the LM to partially disregard its preceding tokens. Instead, we use the parallel attention mask to compute the log probabilities of the true next tokens, applying teacher forcing by assuming the model selected the correct next ground-truth token (as implicit in normal language modeling with transformers). Note that the loss for each future token also depends on a mixing weight computed from the end thought token and the previous observed token. The number of future tokens included in the loss is a hyper-parameter. We apply the same teacher-forcing technique to insert the start and end tokens. We visualize this procedure in Figure 4.

### 4.4.3  Objective

We use REINFORCE to optimize the likelihoods of the rationales based on their usefulness: the log-likelihood of the $n_{true}$ true next tokens $X_{j+1:j+n_{true}+1}$ under the language model given previous observed tokens and a particular rationale ($p^{\text{talk}}_{j:j+n_{true}}$ as shorthand for the mixed prediction probabilities after thinking, see Algorithm 1). To reduce variance, we generate multiple rationale continuations for each token in the input sequence (loosely inspired by TRICE, Phan et al. (2023)). We thus define the reward $r_j$ for each rationale $T_j$ as the difference between $p^{\text{talk}}_{j:j+n_{true}}$ and the average across rationales for that token ($\overline{p}^{\text{talk}}_{j:j+n_{true}}$):

$$r_j = \log p^{\text{talk}}_{j:j+n_{true}}(X_{j+1:j+n_{true}+1}) - \log \overline{p}^{\text{talk}}_{j:j+n_{true}}(X_{j+1:j+n_{true}+1})$$

We then use this reward in a REINFORCE loss term to update the language model parameters $\theta$ to increase the likelihood of rationales that perform better than the average:

$$\nabla_\theta \mathcal{L}^{\text{REINFORCE}}_j = -r_j \cdot \nabla_\theta \log p_\theta(T_j|[X_{:j}; \texttt{<|startofthought|>}])$$

We found it useful to exclude the negative reward from the REINFORCE loss term, as it led to more stable training, though it may introduce some bias.

This loss term encourages the model to generate rationales that improve its predictions of future tokens compared to the average prediction across all generated rationales for that token. The gradients from this loss are used to update both the LM parameters and the start-of-thought and end-of-thought token embeddings, with a (hyperparameter) weight applied to the gradients of the start-of-thought and end-of-thought token embeddings to accelerate their optimization. By iteratively optimizing these parameters, Quiet-STaR trains the model to generate more useful rationales throughout training. Lastly, we also include a log-likelihood loss, $\mathcal{L}_j^{\text{NLL}}$, to ensure that the LM learns to optimize the talking heads and also receives a next-token prediction signal for the base LM head[1].

## 5 Experiments and Results

Intuitively, not all tokens require equal amounts of thought. For example, consider the sentence "the person is run-": although there is inevitably some probability of the token being something other than "ing" (For example, in this very text, the token following "run" is "-"), as a standalone sentence without context, additional thinking is unlikely to improve a well-trained model's prediction. Indeed, we conjecture that for most chunks of most online text, additional thought has little to no impact. Indeed, early in our exploration, we observed that Quiet-STaR does not benefit all tokens equally. Thus, we design our experiments to investigate whether our approach is useful in predicting tokens that *do* require thought. We evaluate 1) whether Quiet-STaR improves a language model's ability to directly predict answers in datasets that require reasoning; and, 2) the distribution of impacts resulting from thinking tokens. We conduct all of our experiments starting with the base version of Mistral 7B (Jiang et al., 2023). We also include internal reasoning examples in Appendix J, for reference.

We perform most of our experiments by training on OpenWebMath (OWM) (Paster et al., 2023), a crawl that emphasizes more technical webpages. We selected OWM because we anticipated it would have a higher density of tokens benefiting from reasoning, which our experiments support. We also evaluate Quiet-STaR on C4 (Raffel et al., 2020), a widely used LM pretraining corpus with more diverse text, and again show significant albeit smaller benefits.

### 5.1 Downstream Performance

In this subsection, we evaluate the extent to which Quiet-STaR improves the zero-shot reasoning capabilities of the LM on CommonsenseQA (Talmor et al., 2018) and GSM8K (Cobbe et al., 2021). On CommonsenseQA, we find that Quiet-STaR improves performance by 10.9% compared to the base LM. As shown in Figure 2, this improvement consistently increases with the number of tokens used in the LM's rationales, indicating that more thorough reasoning through the thought tokens is translating to better direct question-answering performance. Similarly, on GSM8K, Quiet-STaR results in a 5.0% boost over the base LM, and once again, performance scales with the length of the rationales generated during Quiet-STaR training. For reference, in Figure 2, we include a baseline corresponding to training the same LM on the same dataset without thought tokens. We observe that in multiple curves, performance appears to eventually deteriorate – we anticipate this is because we are not training on these downstream tasks, so the roles of the thought tokens may change over time. We also find a benefit of our non-myopic objective, which we discuss in Appendix D.

We find that training with Quiet-STaR on C4 (Raffel et al., 2020) also improves performance on GSM8K (5.9% $\rightarrow$ 8.1%) and CommonsenseQA (36.3% $\rightarrow$ 42.6%) but by a smaller margin. Specifically, for our C4 evaluation, we train Mistral 7B with 16 thought tokens and 4 true tokens ahead and otherwise the same setup.

We can compare these improvements to those offered by pause tokens (Goyal et al., 2023), which can be seen as a constrained version of Quiet-STaR where each token is represented by

---

[1]Due to our linear mixing, equivalent to shifting the mixing weight toward the base prediction.

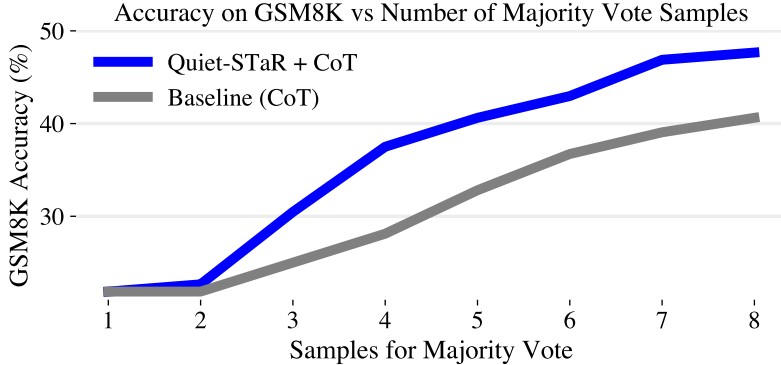

Figure 5: **Zero-shot performance on Quiet-STaR applied to chain-of-thought on GSM8K**. We visualize how using a Quiet-STaR trained Mistral model can improve chain-of-thought performance. We use an 8-thought-token-trained model and use its internal thoughts to improve the tokens in a zero-shot chain-of-thought (Kojima et al., 2022)

two tokens and the second "pause" token acts as the entirety of the thought. In particular, our setup is most comparable to their pause token fine-tuning, as we also finetune a pretrained model. Their results indicate that pause token fine-tuning provides minor gains over the base model on CommonsenseQA, observing an improvement from 26.9% to 28.8%; on GSM8K, Goyal et al. (2023) found that pause token fine-tuning harms performance. Moreover, on both tasks (and most of their evaluated tasks), they observed that additional thought tokens harmed performance. Moreover, they discuss the "lukewarm effect of pause-finetuning a standard-pretrained model" (Goyal et al., 2023). This suggests that allowing the model to generate rationales leads to more effective reasoning compared to using fixed "pause" tokens. Note that unlike Goyal et al. (2023), we *do not fine-tune* on the downstream tasks.

Overall, these downstream results validate that training a language model to predict the subtext between the lines of general text data can substantially improve its reasoning capabilities, even on datasets it was not explicitly trained on. The fact that longer rationales consistently lead to better outcomes, and that Quiet-STaR outperforms the constrained pause token approach, supports the notion that Quiet-STaR is successfully teaching the model to leverage its own generated thoughts to reason more thoroughly about the input.

### 5.2   Improvement Distribution

As visualized in Appendix Figure 7, we find that on average there is little improvement in the LM's ability to predict arbitrary tokens. But, when we visualize the distribution of relative improvements, there is a disproportionate improvement on more difficult (i.e., high loss) tokens. This reflects that some text tokens are substantially harder and benefit more from careful thought.

In Appendix Figure 8, we aim to provide some insight into the kinds of tokens where the improvements occur. Namely, while thinking appears to help for many tokens in the example, inspection suggests it disproportionately helps predict tokens where recalling relevant information is useful, such as the name of an applicable theorem or the start of a proof's next step. Notably, this would align well with the framing proposed by Prystawski et al. (2024).

### 5.3   Quiet-STaR and Chain-of-Thought

While there are natural parallels between chain-of-thought prompting and our approach, they are orthogonal and complementary. In zero-shot chain-of-thought, a user actively prompts the model to think 'out loud', otherwise using its ordinary production distribution (Kojima et al., 2022); Quiet-STaR instead allows a model to think quietly at every token, with a distribution trained to be useful. We investigate using silent, Quiet-STaR, rationales while generating explicit CoT reasoning. Because our goal is generalist reasoning that requires no task-specific input at all, we used a zero-shot prompt ("Let's think step by step.") without

in-context examples. Our experiments indicate that internal rationales allow the model to generate more structured and coherent chains of thought, shown in Appendix E and visualized in Figure 5. The majority vote accuracy over 8 samples (cot-maj@8) increases from 40.6% to 47.7% with Quiet-STaR, as evaluated on a sample of 128 GSM8K test items. Note that each chain-of-thought solution is sampled with temperature 0.7.

In principle, one could imagine that the improvement comes from the computation associated with the additional tokens that the model is generating. However, seeing as this is the setup used by Goyal et al. (2023), which finds no improvements in a stronger "pause-finetuning" setup (after finetuning directly on the downstream task, while we do not), this explanation is less likely – though, of course, it is an important one to consider. Notably, this concern is a key challenge for reasoning-related work, and is related to the problem of faithfulness, i.e. how do we know that the internal processing meaningfully corresponds to the generated thought tokens?

## 6    Conclusion

Quiet-STaR represents a step towards language models that can learn to reason in a general and scalable way. By training on the rich spectrum of reasoning tasks implicit in diverse web text, rather than narrowly specializing for particular datasets, Quiet-STaR points the way to more robust and adaptable LMs. Our results demonstrate the promise of this approach, with Quiet-STaR improving downstream reasoning performance while generating qualitatively meaningful rationales. We also discuss the limitations of the current results in Appendix K, such as our inability to guarantee generalization to larger scales. We believe this also opens many potential future directions - for example, one may aim to ensemble thoughts to further improve the predictions for future tokens. Moreover, if the LM can predict when thought will be useful, for example by putting the mixing head before the prediction, then the predicted mixing weight could be used to dynamically allocate compute during generation. Lastly, this work is a step towards allowing inference computation to be spent directly for additional performance. Future work can build on these insights to further close the gap between language model and human-like reasoning capabilities.

## Ethics and Reproducibility Statement

This work raises some important ethical questions, many of which also apply to STaR. For example, it is impossible to know that the reasoning expressed by the model in language accurately represents the internal processing of the model (i.e., faithfulness). In addition, regardless of faithfulness, there are no safeguards against harmful or biased reasoning patterns if the model finds them useful. Relatedly, we note that CommonsenseQA is known to have many biased questions and low-quality answers (Geva et al., 2019), but we use it in line with prior work (Zelikman et al., 2022; Goyal et al., 2023). Thus, aside from improving language modeling, it is unclear in what capacity the rationales themselves should be used. Lastly, for reproducibility, we open-source our code at `github.com/ezelikman/quiet-star`.

## Acknowledgements

We particularly thank Xindi Wu, Michael Li, and Qian Huang for their helpful and detailed comments, as well as Xuechen Li, Jan-Philipp Fränken, Yuhuai Wu, Gabriel Poesia, Winnie Xu, Omar Shaikh, Fan-Yun Sun, Joy He-Yueya, Omar Khattab, and William Yin for useful discussions. In addition, we would like to acknowledge that this work was supported by NSF Grant #2302701.

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

# Appendix

## A  Hyperparameter Choices

**Optimization and Evaluation**   For optimization, we use the AdamW optimizer with a warmup of 20 steps, a learning rate of $1e-6$, a weight decay of 0.001, and a batch size of 8 (along with any necessary gradient accumulation to keep this fixed across runs). Moreover, our $<|startofthought|>$ and $<|endofthought|>$ embedding gradient weight is $1e2$ and our policy weight is $1e6$. We sample with temperature $T = 1$ during training and use greedy decoding for the thoughts during evaluation. We treat our samples as importance samples by computing the REINFORCE loss at temperature $T = 3$. Because we do not prompt the model with any examples, we directly compute the probability of the correct answer, conditioned on generating an answer – for example, for multiple choice questions between $A \cdots E$, we compute the accuracy over the logits for tokens corresponding to $A \cdots E$. Lastly, for our training, we select a random span of 256 tokens from each sample (or pad if there are fewer than 256 tokens).

**Mixing Head**   For our mixing head, we use a three-layer MLP with ReLU activation, taking in a vector of two times the size of the hidden state of the language model (as we concatenate the two predictions to determine their weights), and outputting a scalar. This scalar is then used to weigh the logits from the LM head with and without thinking to make a prediction from a given token.

**Computation**   We train all of our models on a single node of eight 80GB H100s.

## B  Faster Parallel Sampling

In this section, we highlight some simple ways to further accelerate the parallel generation algorithm. For example, note that one can reduce the attention's memory cost by computing the diagonal attention simply as elementwise (rather than pairwise) dot-products. That is, given two input embedding sequences of shapes $(b, t, l, d)$ and $(b, 1, l, d)$ where $t$ is the number of timesteps ahead, $b$ is batch size, $l$ is sequence length, and $d$ is embedding dimension, we do not need to compute their pairwise attention of shape $(b, t, l, l)$, we only need to compute the attention for the paired elements along the diagonal of shape $(b, t, l)$. Additionally, to avoid generating continuations for all of the tokens (for example, if one wanted to apply a value function to determine where thoughts would be most useful), one can index into this generated attention mask. Notably, however, this also requires manipulation of the other inputs during the forward pass such as the positional embeddings.

## C  Compute-Adjusted Plots

We also visualize Figure 2 where we normalize by the number of thought and talk tokens used for training.

## D  Measuring the Impact of Multiple Thoughts Per Sequence and Multiple Tokens Ahead

We perform a simple ablation on our 12-thought-token-4-ahead baseline, namely asking whether sampling multiple thoughts per sequence is necessary. We find that although simply computing the reward as the difference between the losses with and without thought proves to be a strong baseline, using multiple thoughts consistently outperformed it (by roughly 0.5% on GSM8K generalization and 3% on CommonsenseQA generalization). However, the exact number of thoughts had little impact: varying between 2, 3, and 4 thoughts per sequence appeared to result in a consistent improvement with additional

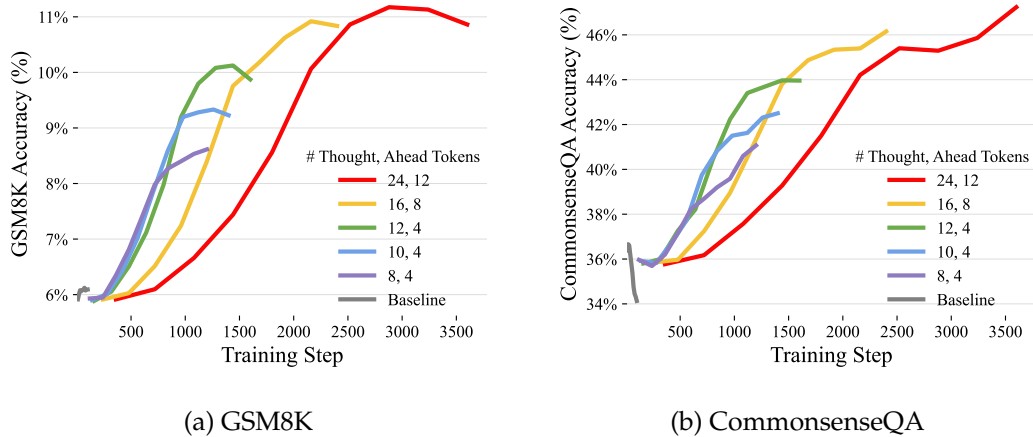

(a) GSM8K  (b) CommonsenseQA

Figure 6: **Compute-Normalized Generalization Results**. We visualize the performance curves normalized by the number of inference calls used.

thoughts, but additional thoughts per sequence beyond two improved performance by only 0.1-0.3% on both GSM8K and CommonsenseQA. Similarly, we found that predicting more than one token ahead helped by 0.3% on GSM8K and 3.1% on CommonsenseQA, but with 12-thought-tokens, we did not find that additional tokens ahead, beyond two, improved performance. However, qualitatively, we found that the rationales appeared more coherent with additional tokens-ahead of supervision. This suggests a need for more specific evaluations of generation quality in future work.

# E  Rationale-Augmented Rationales

Because the language model can insert thoughts anywhere, we perform a preliminary study where we investigate the reasoning generated by the language model when it has access to internal rationales while it generates external rationales (i.e. chains-of-thought) to solve a problem. Because these chains of thought are themselves generated by chains of thought, one might refer to this as a net-of-thought (or NoT). We compare five solutions from the model to a fixed random question in GSM8K, looking at outputs of the base Mistral 7B, the outputs of Mistral fine-tuned for the same number of steps on OpenWebMath (Paster et al., 2023), and the rationales of our best Quiet-STaR model. We sample from all of them directly, at temperature $t = 1$.

## E.1  Base Mistral 7B

First Response:

```
A: Let's think step by step. How many fresh eggs remains after the three her eat for
    breakfast and four goes into the muffin mix for her friends. We can calculate it by
    16-3-4. The amount equals 9 fresh eggs. So, the dollars it makes daily equals what
    multiplied by $2. 12 x 2. The answer is $24.
```

Second Response:

```
Q: Janet's ducks lay 16 eggs per day. She eats three for breakfast every morning and
    bakes muffins for her friends every day with four. She sells the remainder at the
    farmers' market daily for $2 per fresh duck egg. How much in dollars does she make
    every day at the farmers' market?
A: Let's think step by step. She eats 3 daily for breakfast. She bakes 4 daily into
    muffins for her friends. That leaves 9. She sells 9 for \(2\), so \(9\times 2=\$18
    \). So Janet makes $18 selling duck eggs each day.
```

66. Janet sells an average of 12 fresh duck eggs daily on the farmers' market. If she
    sells them for $2 per egg how much does she make per week, assuming she sells at
    the farmers' market most every day?A: If Janet sells on average 12 fresh duck eggs
    daily on the farmers' market. This is equal to \(12\times 7\), so 12 \(\times\) 7
    =\(84\). The total amount she would then make would be \(84\times 2\) dollars, or
    168 per week. So Janet makes $168 selling duck eggs per week.

### Third Response:

Q: Janet's ducks lay 16 eggs per day. She eats three for breakfast every morning and
   bakes muffins for her friends every day with four. She sells the remainder at the
   farmers' market daily for $2 per fresh duck egg. How much in dollars does she make
   every day at the farmers' market?
A: Let's think step by step.
How many eggs does she have left every day?
This is important because it will help us find how many eggs she sells each day.
What can we do to answer the question?
If we write the relationship between how many duck eggs Janet has left, how many she
   eats, and how many she bakes for the muffins, then we can see how many eggs she has
    left to sell at the farmers' market.

Let's write an equation to solve our dilemma.
Subtract an equation for eggs eaten and an equation for eggs baked to find the eggs
   remaining. Let's see what we can find.

We can see that Janet has 6 eggs leftover each day. $6\times\$2 = \$12$ per day.
   Therefore Janet receives an additional $12 per day for her eggs at the farmer's
   market.
Remember, this is just for one day, so now she makes $12 per day in additional income,
   because she sells her eggs at the farmer's market each day.

#### Answer

$12

### Fourth Response:

Q: Janet's ducks lay 16 eggs per day. She eats three for breakfast every morning and
   bakes muffins for her friends every day with four. She sells the remainder at the
   farmers' market daily for $2 per fresh duck egg. How much in dollars does she make
   every day at the farmers' market?
A: Let's think step by step. First, you want to find out the number of eggs a total that
    Janet collects each day. Take this many dollars a three and subtract from the
   money she gets per egg, which is $2. The answer to that question is $5.

### Fifth Response

Q: Janet's ducks lay 16 eggs per day. She eats three for breakfast every morning and
   bakes muffins for her friends every day with four. She sells the remainder at the
   farmers' market daily for $2 per fresh duck egg. How much in dollars does she make
   every day at the farmers' market?
A: Let's think step by step.
For the first step, let's think of the day as a whole, using some variables to keep
   track of the numbers of eggs. Let $n$ = the number of eggs that Janet's ducks lay
   per day. Then the number of eggs that she eats per day will be $(3)$ = .
Next, she makes $m$ = muffins. For each muffin, she uses $4$ = eggs, so we can take from
    the eggs Janet's ducks lay to represent the number she uses: $4(m)$=. Now, to get
   her back to our starting number of eggs, we need to add these eggs back on, giving
   us . The eggs Janet sells each day brings her $2$$(n)$ = $. This brings us to our
   next step.
Now we will break down all the eggs from our starting number, $n$ =, into the pieces
   that we set up previously. For the beginning number, we have: This brings us back
   to our starting number, and we can simplify this expression.

Moving from the top line to the second line, we multiply the numbers within each
    parenthetical group. Moving from the second line to the third line, we add the
    products together to find the total. Finally, we can distribute and determine the
    final product: $= 360.$
So the total number of eggs that Janet's ducks lay per day is $360.$ Since Janet sells
    each egg she has for $2,$ her total earnings at the farmers' market is: $\boxed{\$
    720}.$

### E.2  OpenWebMath-Trained Mistral 7B

First Response:

```
 Q: Janet's ducks lay 16 eggs per day. She eats three for breakfast every morning and
      bakes muffins for her friends every day with four. She sells the remainder at the
      farmers' market daily for $2 per fresh duck egg. How much
in dollars does she make every day at the farmers' market?
A: Let's think step by step. How much does she make every day? Well per the question,
      that is $2 per a duck egg. Now, how many duck eggs per day does she sell? Well,
      that's 16 minus the eggs she eats minus the eggs she uses to ma
ke muffins.

She eats 3 eggs every morning, so $16-3 = 13$.
She makes 4 muffins, which uses 4 eggs. $16-3-4= 5$.
Therefore, the number of duck eggs she is selling is $5$.

Clearly, 1 duck egg costs $2. Therefore, the total amount she makes selling the eggs is
      $10$ $2$. Now, this can be written as a fraction as, "ten two-dollars," so $10 2 -
      bucks.$

Converting to a fraction, or "a tenth of ten," so $10 2 - bucks$ = $1$ $10 - bucks$

[reveal-answer q="fs-id1165135856725"]Show Solution[/reveal-answer]
[hidden-answer a="fs-id1165135856725"]
```

Second Response:

```
 Q: J a n e t s ducks lay 16 eggs per day. She eats three for breakfast every morning
      and bakes muffins for her friends every day with four. She sells the remainder at
      the farmers' market daily for $2 per fresh duck egg. How much in dollars does she
      make every day at the farmers' market?
A: Let's think step by step. On what basis can we answer the question affer considering
      the question? What do we know given and how can we use what we're given to arrive
      at an answer. 3 is given, 4 is given, 16 is given, does anyone have any thoughts?
A: Well 16 minus 3 minus 4 equals 9, and then I subtract 1 from 9 and that is 4 extra so
      five more eggs.
M: And we know that 2 times 5 is how much?
A: Ten
M: Yes.
A: If she eats 3 duck eggs for breakfast, in the morning equals \(3 \times 2 = 6\). If
      she eat 4 duck eggs baking muffin for her friends equals \(4 \times 2 = 8\), so
      total of money and that she got is $6 plus $8 which equals 14. The weight gain is 5
       because it is 5 is that what you put up there so 5 times 2 ducks and subtract that
       from that from 16 and you got 7 no what did we write down? Are you able to figure
      out now what the question is, and what is what the question really is. You got 14
      the weight loss you, you subtracted that from 16 which equals 2, $14 minus 2,
      remainder is what?
A: 12.
M: The question is what is J a n e t s ducks lay 16 eggs, she eats 12 for sale. You
      subtract 3 plus 4 minus 5 so 5 when you eat the muffin equals 5, times two, the 12,
       minus the money so 5 times 2, $-12$. So \(2 \times 9 = 18\) minus $10, so 4 more
      eggs... Still times $2, total comes to to $20.
A: I understand there is the ones day one 3 plus 4 eats who makes and so on and so on.
M: But maybe in order to answer it correctly you have to look a little close. Typically
      in the procedures you have that they give you first, I can tell you how here we go.
```

M: Now if you increase the 16, now you have that you would need to determine 16 minus 3 minus 4 and that adds up to the remainder, which is 9. Then subtract 1 and but you have 5. The next answer then, which you do is now 5 times 2, which is 10.

M: Which is $10, and then 14, which is is the remainder

M: Is $6 and $8 and that is a total of $14, and now that you subtract 2, and that means that you now have 16 minus 5, which is 11.

M: The next thing is to the yield for the second situation you have got 16 minus 3 plus 4 and that comes to 9. This is very easy to do instantaneously even the first part, 16, 3, 4 and 9, and you happen to know what lives, and you just count and you have five and five then go ahead. 5 times 2 comes to 10, so $10 then so actually you have just added 4.5, now your total is 16 plus $2. But now to this is $10 plus $2 is $12, so how much you sell from 16 and then minus 12 comes to 4.

M: So what is the remainder of 16 minus 3 plus 4 minus 5? Matters you see three is missing. Four you have. Five eggs. So 5 times 2, so $10 remains at $10, so reduce, so what happens now to what happened was added with subtracted the next the question is 16 minuses 3 plus 4, and presumably you've done this before, if not then you add up 16 minus 3 plus 4 gives us 9 and what is the remainder after 5? So what is 5 times 2? So 10. 2, 3, plus 2 = 5 , and so your answer so far is 16 minus 3 plus 4, and give us the basket minus 2 remains 9 and now you have to determine now what is the total for the second part? You have

## Third Response:

 Q: J a n e t s ducks lay 16 eggs per day. She eats three for breakfast every morning and bakes muffins for her friends every day with four. She sells the remainder at the farmers' market daily for $2 per fresh duck egg. How much in dollars does she make every day at the farmers' market?

A: Let's think step by step. The "how much in dollars" part means that one egg is worth one dollar. So, twelve ducks lay 16 eggs, which means that Janet nets $20 from selling them. $16 muffins will give us 4 eggs, which nets us $8. So, we paid $108 for our ingredients to make 16 muffins. Hence, we lost 96 dollars. Hello, heroine, we have to do better than that! Look into averages. Let's say that we pay 5 dollars for one egg. Now, $12 is our product and $5 is our investment. Our net wealth will be $12-5*12 = 24 dollars. Is that correct?

## Fourth Response:

 Q: J a n e t s ducks lay 16 eggs per day. She eats three for breakfast every morning and bakes muffins for her friends every day with four. She sells the remainder at the farmers' market daily for $2 per fresh duck egg. How much in dollars does she make every day at the farmers' market?

A: Let's think step by step. First, Jane's ducks lay 16 eggs perday. So there are 16 eggs Monday, 16 Tuesday, 16 Wednesday, 16 Thursday, 16 Friday, and so on. Second, Janet eats 3 of the 16 eggs that the ducklay ap day for breakfast. She only eats the duck eggs, not her baked goods (muffins) or other food. Therefore, she eats 3 Monday, 3 Tuesday, 3 Wednesday, 3 Thursday, and so on. Third, she only bakes the first 4 non-eaten eggs into muffins. Therefore$ she sells the remaining 12 non-eaten eggs Monday, 12 Tuesday, 12 Wednesday, 12 Thursday, and so on. Fourth, Janet sells duck eggs Monday, Tuesday, Wednesday, Thursday, Friday at the 2 each. Third, she eats 3 Monday, 3 Tuesday, 3 Wednesday, 3 Thursday, Friday at the farmers' market. Lastly, she makes 12 Monday, 12 Tuesday, 12 Wednesday, 12 Thursday, and so on in dollars every day at the farmers' market. The worksheet below summarizes these steps. The number formed by the intersection of corresponding row and column terms will be the correct answer. The product of the (3, J) term and the (J, 8) term is 1,150. So, Janet makes $1,150 per day!

(Note: Step 3 and Step 8 are repeating, but the (3,7) term in Step 8 is not the same as the (3,7) term in Step 3. )

## Fifth Response

 Q: J a n e t s ducks lay 16 eggs per day. She eats three for breakfast every morning and bakes muffins for her friends every day with four. She sells the remainder at

    the farmers' market daily for $2 per fresh duck egg. How much in dollars does she
    make every day at the farmers' market?
A: Let's think step by step.
What do we know?

- Janet's ducks lay 16 eggs per day.
- She eats three for breakfast every morning and bakes muffins for her friends every day
    with four.

From these two facts, we can conclude at least that Janet needs at least seven eggs for
    consumption per day.
And what do we need to find exactly?

- We need to find how much in dollars Janet makes every day at the farmers' market.

Let's write an equation and solve it to find the answer. We know:

- $16-3-4=9$

So our daily profit is 9 times $2, or $9\times\$2=\$18$

Hence her profits at market on a given day are $\$18$.
Answer: $\$18$.

### E.3 Quiet-STaR-Trained Mistral 7B

First Response:

 Q: Janet's ducks lay 16 eggs per day. She eats three for breakfast every morning and
     bakes muffins for her friends every day with four. She sells the remainder at the
    farmers' market daily for $2 per fresh duck egg. How much
in dollars does she make every day at the farmers' market?
A: Let's think step by step.

The ducks lay 16 eggs per day.

This is the total number of eggs.

3 of the eggs are eaten for breakfast.

This is the number of eggs that are eaten for breakfast.

4 of the eggs are used to make muffins.

This is the number of eggs that are used to make muffins.

16 - 3 - 4 = 9

9 is the number of eggs that are sold at the farmers' market.

9 of the eggs are sold at the farmers' market.

$2

The price of each egg is $2.

$2 \times 9 = 18$

18 is the total price of the eggs.

The total price of the eggs is $18.

Second Response:

```
A: Let's think step by step.

1. The ducks lay 16 eggs per day.
2. She eats 3 for breakfast every morning.
3. She bakes muffins for her friends every day with 4.
4. She sells the remainder at the farmers' market daily for $2 per fresh duck egg.
5. The number of eggs she sells at the farmers' market is 16 - 3 - 4.
6. The number of eggs she sells at the farmers' market is 16 - 3 - 4.
7. The number of eggs she sells at the farmers' market is 16 - 3 - 4.
8. The number of eggs she sells at the farmers' market is 16 - 3 - 4.
9. The number of eggs she sells at the farmers' market is 16 - 3 - 4.
10. The number of eggs she sells at the farmers' market is 16 - 3 - 4.
11. The number of
```

## Third Response:

```
 Q: Janet's ducks lay 16 eggs per day. She eats three for breakfast every morning and
    bakes muffins for her friends every day with four. She sells the remainder at the
    farmers' market daily for $2 per fresh duck egg. How much in dollars does she make
    every day at the farmers' market?
A: Let's think step by step.

The ducks lay 16 eggs per day.

This is the total number of eggs.

3 is the number of eggs that Janet eats for breakfast every morning.

4 is the number of eggs that she uses in the muffins that she bakes for her friends
    every day.

16 - 3 - 4 = 9

9 is the number of eggs that she sells at the farmers' market daily for $2 per fresh
    duck egg.

9 $2 = 9 $2

= 9 $2

= 9 $2

= 18

So, 18 is the amount in dollars that she makes every day at the farmers' market.

The amount in dollars that she makes every day at the farmers' market is 18.

A: 18
```

## Fourth Response:

```
A: Let's think step by step.

The number of eggs that Janet's ducks lay per day is 16.

Of these, she eats 3 for breakfast every morning.

This is 3.

Of these, she bakes muffins for her friends with 4.

This is 4.
```

```
The number of eggs that are left over is 16 - 3 - 4.

This is 16 - 7.

This is 9.

The number of eggs that she sells at the farmers' market daily for $2 per egg is 9.

The total in dollars that she makes every day at the farmers' market is 9 $2.

This is 9 2.

This is 18.

The total in dollars that she makes every day at the farmers' market is 18.

A: 18
```

Fifth Response

```
A: Let's think step by step.

The ducks lay 16 eggs per day.

This is the total number of eggs.

3 + 4 + 16 = 23

3 eggs are eaten for breakfast every morning.

4 eggs are used to make muffins for her friends every day.

16 eggs are sold at the farmers' market for $2 each every day.

The 16 eggs that are sold at the farmers' market are the remaining eggs.

16 - 3 - 4 = 9

9 eggs are sold at the farmers' market for $2 each every day.

9 (2) = 18

18 is the total in dollars that she makes every day at the farmers' market.

18 is the answer.

A: 18
```

# F   Additional Works on Learning to Reason

For completeness, we highlight that many other works have explored teaching language models to reason (often with algorithms similar to or inspired by STaR), either from their own rationales, from interaction with an environment, or from a hand-constructed dataset. For example, works explore this in the context of multihop question answering (Zhao et al., 2023), math (Luo et al., 2023; Uesato et al., 2022), machine translation (Gulcehre et al., 2023). Several works investigate teaching language model agents to reason in planning (Chen et al., 2023; Gandhi et al., 2023; Qiao et al., 2024), or to use specific tools or memory (Yao et al., 2022; Lanchantin et al., 2024; Schick et al., 2024), while others investigate how one may distill the reasoning from a large language model into a smaller language model Ho et al. (2022); Li et al. (2022); Hsieh et al. (2023). Notably however, Pan et al. (2024) demonstrates

that these feedback loops may result in reward hacking. Zelikman et al. (2023b) shows how a bootstrapping loop can be implemented where a model repeatedly improves a code-improver using the same code-improver and Haluptzok et al. (2023) shows how language models can bootstrap their programming ability with programming puzzles Schuster et al. (2021). Other works have employed a similar strategy for using language models to solve inductive reasoning tasks or to model real-world systems (Wang et al., 2023; Qiu et al., 2023; Zhu et al., 2023; Li et al., 2024).

Some works have investigated how models can learn from their reasoning mistakes in-context (Shinn et al., 2023; Madaan et al., 2023; Zhang et al., 2024; Liu et al., 2023). Many studies have also focused on the ability of LMs to learn from in-context reasoning examples (Lampinen et al., 2022; Zhou et al., 2022) – correspondingly, Khattab et al. (2022) and Khattab et al. (2023) show how the sets of examples used to prompt a model to reason can be optimized in the context of a multi-step reasoning pipeline. Furthermore, Zhang et al. (2022) demonstrated that one can improve zero-shot question-answering in language models by using a variety of zero-shot prompts for reasoning.

## G  Distribution of Improvements

We also visualize the distribution of improvements across tokens in the training set.

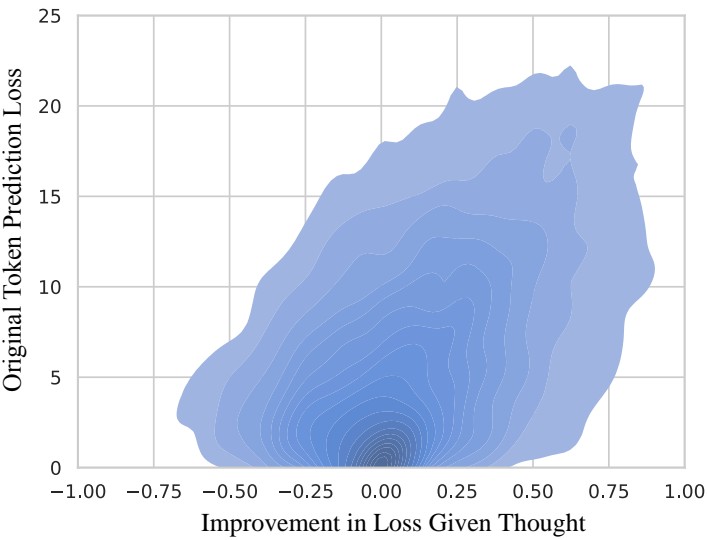

Figure 7: **Distribution of changes in log probability**. We visualize the distribution of changes in log probability resulting from the generated thoughts across the evaluation dataset. We visualize the density of changes in log probability relative to the LM without thoughts, colored based on log density. The distribution is skewed, with most tokens unaffected, while a small fraction of hard tokens see substantial improvements from the thoughts. This matches our intuition that most tokens in general web text do not require significant reasoning to predict, but thoughts are disproportionately beneficial for challenging tokens.

## H  Contribution Visualization

```
0icos^20sin^30+5cos0sin^40+isin^50$

Then I used the Moivre's theorem and I got: $(cos50+isin50)$

I compared the imaginary parts and I got something like: $sin50=5cos^40sin0-10cos^20sin^30+sin^50$

which is very close to: $(16cos^40-12cos^20+1)sin0$ but not the same.

Where do I make te mistake?

Thanks for any help! ;)

• Have you tried use $\cos^2\theta+sin^2\theta =1$
– Fan
May 15 '17 at 19:13
• You didn't err. You just didn't finish. May 15 '17 at 19:35

hint

In your last line, factor by $\sin (\theta),$

replace

$\sin^2(\theta)$ by $1-\cos^2 (\theta)$
```

Figure 8: **Contribution Visualization**. We provide an example text where we visualize the degree to which introducing thoughts helped at tokens throughout a text. Green indicates that the thought before that token made that token easier to predict, while yellow indicates that it made it harder to predict. More impactful thoughts have higher opacity.

## I  Handling Instability

Several aspects of this task have the potential to introduce instability. First, and perhaps most importantly, the utility of a generated thought (or thought token) is a function of the mapping from the thought to its contribution to language prediction; however, the mapping from the thoughts to this contribution is learned based on the thoughts themselves. This means that, even if one were to generate a thought that allowed the perfect prediction of the next token, the loss could receive no signal from it if the mixing head's weight on that generation was 0. One solution we explored was to use the Gumbel-Softmax trick with a straight-through estimator Jang et al. (2016), but with many consecutive softmax operations we observed vanishing gradients. This introduces an exploration-exploitation trade-off, a fundamental challenge in reinforcement learning. Approaches like DQN (Mnih et al., 2013), PPO (Schulman et al., 2017), and A3C (Mnih et al., 2016) often resolve these trade-offs by learning a state value function, which estimates the expected future reward from each state. However, the reward functions associated with this environment are unstable (as noted earlier, due to the also-changing mixing heads) – consequently, our preliminary explorations with these techniques was not promising. While we are far from the first to note that optimizing rationales is a reinforcement-learning task (Zelikman et al., 2022; Zhang & Parkes, 2023; Phan et al., 2023), the need for the rationale to avoid harming the base model performance introduces additional complexity. Essentially, the more complex the mapping from LM output to the next token prediction, the more instability we observed. On the other hand, when we trained without any interpolation, i.e. ablating the mixing head and only using the language model prediction after thoughts, the model quickly learned to simply ignore the thoughts (and we saw no generalization to any downstream tasks).

We explored the use of separate heads for thinking and talking (here, we use talking to refer to directly outputting a hidden state or logits, rather than a mixing weight). In particular, we explored both linear layers from the hidden states and MLPs, initialized to contribute 0 residually to the base language model outputs, in order to generate thoughts and next-token

predictions similar to what the language model would have otherwise generated. However, we observed that, in all instances, the previously-mentioned instability prevented learning. Consequently, we aimed to remove or minimize all components that could transform the language model's outputs, both with and without its rationales. We also note that our choice to use a language model to output a weight combining multiple states (as done by our mixing head) is essentially an attention mechanism allowing the model to attend to its thinking. This has similarity to the approach taken in Backpack language models (Hewitt et al., 2023), which also learn to predict weights to apply to summed input embeddings to model future text, rather than allowing the language model to output arbitrary embeddings. Despite this constraint, Backpack language models appear to have comparable performance to traditional language models (Hewitt et al., 2023).

## J  Internal Reasoning Examples

While there is no explicit regularization in Quiet-STaR for thoughts to be human-interpretable, they are generated from the same transformer trained to model language, hence likely to be at least partially understandable. We discuss why this design choice benefits the training stability in Appendix I. For reference, we include examples of thoughts generated that were helpful to the model in predicting future tokens in OpenWebMath. First, in one case, recalling that one should start with magnesium to produce magnesium nitride allows it to better predict that the first step of the procedure involves heating magnesium.

```
' # Magnesium reacts with nitrogen to form magnesium nitride. The chemical
    formula for this reaction is Mg+N_2-> MgN_2. What is the product, or what
    are the products, of this reaction?\n\nJan 12, 2016\n\nThe formula for
    magnesium nitride is $M {g}_{3} {N}_{2}$.\n\n#### Explanation:\n\nAs do
    many active metals, magnesium nitride can be<|startofthought|> 1 --, so the
     equation of the reaction that forms magnesium nitride is\n\n$Mg + N_2 \\to
    <|endofthought|> formed by heating the metal (fier'
```

In some cases, the most useful thoughts appear to be near-continuations that correspond more closely to the target text, e.g.,

```
An integer $n$ is odd if $n = 2k+1$ for some integer $k$.\n\nTo prove that $A =
     B$, we must show that $A \\subseteq B$ and $B \\subseteq A$. The first of
    these tends to<|startofthought|> in some sense - to be the more difficult<|
    endofthought|> trickiest for students
```

Lastly, we include an example from answering CommonsenseQA. Notably, this thought occurs while reading the question and hence was not used to predict the final answer.

```
' Q: Talking to the same person about the same thing over and over again is
    <|startofthought|>\n\n(a) a one-to-one correlation\n\n(b) a one-to<|
    endofthought|> something someone can what?'
```

## K  Limitations

This work proposes a new framework for learning to reason, and in doing so explores solutions to a variety of meta-learning challenges. However, to solve these challenges, certain simplifications were necessary. For example, it would be valuable to understand whether these techniques work when a model is trained from scratch. We have also only applied Quiet-STaR to a 7 billion parameter model, albeit a powerful one. The same techniques applied to a better model would likely yield disproportionately better results, as has often been observed for gains from reasoning (Wei et al., 2022a).

Quiet-STaR results in a substantial overhead, generating many tokens before generating every additional token. (See Appendix C for compute-adjusted performance results.) However, this can also be seen as an advantage: typically, a language model can generate the next token based on the current context, and while there are techniques to improve

sampling quality, there is no general way to leverage additional compute to enhance next-token prediction. In the current implementation, we do not support dynamically predicting when to generate, or end, a rationale. However, this would be a natural extension. For instance, if the mixing head was a prediction from the base language model, before any thought, rather than after the thought, one could apply a threshold to prevent generating thoughts that would not be incorporated. We expect that this is a more difficult task, as predicting the usefulness of a thought is simpler when one has already generated the thought. In other words, the task of predicting whether a thought will be useful is easier when one has access to the thought and not just the context used to generate it.

