# OpenReview forum: "Quiet-STaR: Language Models Can Teach Themselves to Think Before Speaking"
_colmweb.org/COLM/2024/Conference — COLM_

### Official Review · Reviewer_PS48 · 2024-04-22

**Rating:** 8
**Confidence:** 4
**Ethics Flag:** 1

**Summary:**

This paper proposes Quiet-STaR, which trains the model to generate (pseudo-)internal thoughts for predicting every token on the general language modeling task.
Despite the absence of finetuning on downstream tasks/datasets, the model trained by Quiet-StaR achieves significantly better performance on two reasoning datasets (GSM8K and CommonsenseQA) than the baseline model does.

**Reasons To Accept:**

+ I personally believe the general topic of “generating internal thought” is worth researching and this work is a great effort under this topic. Compared to previous works, the proposed method itself is technically good.
    + Quiet-STaR doesn’t rely on randomness (e.g., Goyal et al. (2023) insert pseudo tokens at random locations).
    + It is also used in an unsupervised fashion (e.g., unlike Deng et al. (2023), Quiet-STar doesn’t need any external supervision of the internal thoughts or even its locations) so it is quite scalable.
+ Quiet-STaR is very effective given the significant performance improvement on downstream reasoning tasks despite the model not training on the downstream tasks.
+ It is great to show the generated internal thoughts may contain meaningful reasoning. This indicates that Quiet-STaR does some effective and could-be-interpretable learning. I also would be happy to see more research on that in the future, and it may contribute to some interpretability research.
+ The analysis of improvement distribution is insightful. It is great to empirically show that some tokens need more computation than others do, a hypothesis that is mentioned by many works on “adaptive computation” (I personally still love to see some strong empirical evidence tho it is quite intuitively reasonable).

### Reference
Yuntian Deng, Kiran Prasad, Roland Fernandez, Paul Smolensky, Vishrav Chaudhary, and Stuart Shieber. Implicit chain of thought reasoning via knowledge distillation. *arXiv preprint arXiv:2311.01460, 2023.*

Sachin Goyal, Ziwei Ji, Ankit Singh Rawat, Aditya Krishna Menon, Sanjiv Kumar, and Vaishnavh Nagarajan. Think before you speak: Training language models with pause tokens. *arXiv preprint arXiv:2310.02226, 2023.*

**Reasons To Reject:**

+ As Quiet-STaR incurs significant extra computation overhead, would it be possible to achieve similar performance by spending the extra computation on scaling up the model size & training data (or even just use a larger model during the inference, noting that the extra overhead during inference is also significant)? It is okay even if the answer is YES as I think this work’s value is more about performance improvement, but now the paper lacks such an analysis.
+ [**Not a major concern and could be regarded as a discussion on future work**] I doubt the argument that `We expect that this is a more difficult task, as predicting the usefulness of a thought is simpler when one has already generated the thought`, which was used to justify Quiet-STaR generating many tokens before generating **every** additional token.
    + Some other works on “adaptive computation (i.e., the model spending different computations on different tokens)” succeed in doing this kind of computation prediction ahead of the real computation. For example, Raposo et al. (2024) predicted whether a token needs to pass a layer or not ahead (tho this work doesn't use the technique of pseudo-tokens/internal thought, it is still about computation allocation so I mention it here).
    + Also, thinking about humans' thinking process, we are able to recognize some tokens (e.g., the result of an addition problem) are more difficult than others (e.g., I am sing-*ing*).
+ [**Not a major concern**] I am curious about how the internal thoughts (for difficult tokens) are changed during the training process. Intuitively, the generated thought may make little sense at the beginning, and finally, it can contain some meaningful reasoning as the paper has already shown. This analysis of training dynamics could provide insights into how the REINFORCE loss term works.

### Reference
David Raposo, Sam Ritter, Blake Richards, Timothy Lillicrap, Peter Conway Humphreys, and Adam Santoro. Mixture-of-Depths: Dynamically allocating compute in transformer-based language models. *arXiv preprint arXiv:2404.02258, 2024.*

---

> ### Author Rebuttal · Authors · 2024-05-28
>
> Thank you for the supportive and thought-provoking comments and questions!
>
> > As Quiet-STaR incurs significant extra computation overhead, would it be possible to achieve similar performance by spending the extra computation on scaling up the model size & training data (or even just use a larger model during the inference, noting that the extra overhead during inference is also significant)?
>
> At least in the current framing, Quiet-STaR is presented largely as a continual training / post-training technique, so most likely the best comparison is compute-equivalent training without new parameters. The most expensive runs in our work have compute comparable to training on tens of millions of tokens, roughly a millionth of the pretraining compute (supposing Mistral 7b was trained on trillions of tokens). In contrast, continued training on OWM / C4 results in little improvement (or in some of our baselines, actually results in deteriorations – we’ll also update the figure to make it clearer that our baseline corresponds to naive continual pretraining).
>
> As for inference cost, the question is definitely trickier. LM scaling laws suggest that a sufficiently large model trained on sufficiently much data should likely perform as well as Quiet-STaR, even on difficult tokens. However, it’s not clear how we might estimate the potential size of such a model or its training corpus. Perhaps the simplest answer is that, given a (sufficiently large) pretrained model, Quiet-STaR allows us to trade inference compute for additional performance in a general way (with other recent works like Rapaso 2024 also being exciting steps in this direction). Indeed, we hope this lays the groundwork for more adaptive computation strategies.
>
> > I doubt the argument that "We expect that this is a more difficult task, as predicting the usefulness of a thought is simpler when one has already generated the thought," which was used to justify Quiet-STaR generating many tokens before generating every additional token.
>
> We absolutely agree! We mention this in the conclusion and limitations but dynamically predicting the amount of thinking tokens to generate is an essential future direction. This line is mostly saying that predicting the utility of a generated thought is easier when one has the generated thought (indeed, this must be true if there is variance in the utility of thoughts and a thought provides some information about its usefulness). We’ll clarify this in the revision.

---

### Official Review · Reviewer_vjbp · 2024-05-12

**Rating:** 7
**Confidence:** 2
**Ethics Flag:** 1

**Summary:**

The paper introduces Quiet-STaR, a novel framework that extends the capabilities of Self-Taught Reasoner (STaR) to enable language models (LMs) to infer unstated rationales in arbitrary text. It tackles challenges such as computational costs, lack of initial knowledge on generating internal thoughts, and the need to predict beyond individual tokens. By employing tokenwise parallel sampling, learnable tokens indicating the start and end of thoughts, and extended teacher-forcing techniques, Quiet-STaR facilitates the generation of rationales. Notably, it significantly improves the LM's ability to predict difficult tokens and answer challenging questions directly. Through continued pretraining on internet text corpora, Quiet-STaR achieves substantial zero-shot improvements on tasks like GSM8K and CommonsenseQA, while enhancing the LM's performance in understanding natural text. Overall, this line of research holds significant promise and is likely to draw increasing interest and participation.

**Reasons To Accept:**

* The paper pioneers the exploration of training language models (LMs) to reason comprehensively from text, marking a significant advancement in natural language understanding. Through this pioneering approach, the study not only broadens the scope of LM capabilities but also showcases substantial enhancements in performance across various downstream tasks. This pioneering aspect highlights the paper's contribution to pushing the boundaries of LM-based reasoning.
* The authors propose several innovative techniques, including a parallel sampling algorithm, meta-tokens, and a mixing head, to address challenges encountered during training. These techniques represent cutting-edge solutions that enhance the scalability, efficiency, and effectiveness of the training process.

**Reasons To Reject:**

N/A

---

> ### Author Rebuttal · Authors · 2024-05-28
>
> Thank you for your encouraging, thorough, and positive feedback!

---

### Official Review · Reviewer_fwta · 2024-05-12

**Rating:** 7
**Confidence:** 4
**Ethics Flag:** 1

**Summary:**

This paper introduces Quiet-STaR, an extension of the previous framework STaR, which enhances Language Models (LMs) by enabling them to generate rationales at each token to explain future text, thus improving prediction accuracy. Quiet-STaR leverages the LM's inherent reasoning abilities to generate rationales and employs a REINFORCE-based reward mechanism to train the LM, resulting in improved rationale generation. Notably, Quiet-STaR demonstrates promising potential in significantly enhancing the zero-shot performance of Mistral 7B on two reasoning tasks, CommonsenseQA and GSM8K.

Despite the novel approach and promising results, this paper could benefit from improved writing, clarity, and more comprehensive experiments with diverse models and tasks and competitive baseline methods.

**Questions To Authors:**

-	Instead of generating a rationale for each single token, how about sampling thought based on a chunk of tokens? Would this sampled thought be more meaningful?
-	In Figure 1, the second sampled thought seems like Thought_5. Why does it appear as Thought_6?

**Reasons To Accept:**

+ The idea of Quiet-STaR, which enables LM to learn to reason from diverse text, is novel and interesting.
+ This paper introduces several innovative designs, including a parallel sampling algorithm for efficient generation, customized meta-tokens for marking each rationale’s start and end, and a mixing head for controlling the next-token prediction from a given thought. These contributions have the potential to benefit future research in this area.
+ The experimental results are promising, with Quiet-STaR demonstrating notable performance improvements of 10.9% on CommonsenseQA and 5.0% on GSM8K.

**Reasons To Reject:**

-	The idea of generating a rationale for each token may not be reasonable, as the rationale for a single token may not be contextually meaningful.
-	Figures 1, 3, and 4 are not well explained in the context, causing difficulty in understanding the key components of Quiet-STaR. For example, it is hard to understand the parallel generation process based on Figure 3 and the sketchy description in Section 4.2. It would be nice to have a comprehensive framework of Quiet-STaR and a detailed explanation for each component.
-	Experiments are relatively weak. Only one base model (Mistral 7B) was evaluated on two reasoning tasks. Only one baseline method (Chain-of-Thought) was compared. It would be better to test with other models (e.g., Llama 2 or 3) in a diverse task setting and compare with more competitive baseline methods such as Self-improvement (Huang et al., 2022) and V-STaR (Hosseini et al., 2024).

---

> ### Author Rebuttal · Authors · 2024-05-28
>
> Thanks for the great questions and many encouraging words!
>
> First, to the question about baselines, specifically Self-improvement (Huang et al., 2022) and V-STaR (Hosseini et al., 2024): while these are both excellent papers, they aim to solve a very different problem. Fundamentally, these techniques aim to train a language model to perform well on a given question-answer dataset, specifically by optimizing on questions from that dataset. Instead, our goal is more general: can a language model learn to generate rationales that help predict arbitrary future text?
>
> Both are excellent and inspiring papers. Notably, Huang et al 2022 differentiates itself from STaR primarily by using majority vote instead of ground truth answers, but does not claim to outperform STaR given ground truth – in Quiet-STaR, the continuation of the text is our ground truth. As the length of predicted future text increases, the chance of multiple predictions matching also decreases, so using majority vote becomes impractical. V-STaR is also excellent (note, preprinted the month before the COLM deadline) but for us the future text is the ground truth so the verifier would just be the LM.
>
> > The idea of generating a rationale for each token may not be reasonable, as the rationale for a single token may not be contextually meaningful… Instead of generating a rationale for each single token, how about sampling thought based on a chunk of tokens?
>
> Absolutely, using only a single token ahead would result in a much noisier reward signal – this is why we use the multi-token ahead teacher-forcing trick (see Section 4.4.2).
> > Figures 1, 3, and 4 are not well explained… For example, it is hard to understand the parallel generation process based on Figure 3 and the sketchy description in Section 4.2.
>
> Thank you for this point! We would be happy to add additional explanations for these figures using the additional page for the final version. Do you have a particular change in mind for Figure 3? We tried to directly visualize the attentions of the thought tokens in both parts of the figure.
> > In Figure 1, the second sampled thought seems like Thought_5. Why does it appear as Thought_6?
>
> The second sampled thought shows an example of a thought that would not help with predicting “4”, so the model should not generate thoughts like that.

---

> > ### Comment · Reviewer_fwta · 2024-06-03
> > **Thank you for your response**
> >
> > I appreciate the authors' response and acknowledge the technical merit of this paper. I have increased my score (6 -> 7).
> >
> > Regarding Figure 3, it would be nice to visualize the attentions of the thought tokens. More explanations of the parallel generation in section 4.2 will help. I suggest using the same example from Figure 1 to demonstrate the parallel generation process in Figure 3. Additionally, consider separating the Parallel Inference Mask (right part of Figure 3) into multiple steps.

---

### Official Review · Reviewer_G2Zv · 2024-05-21

**Rating:** 6
**Confidence:** 4
**Ethics Flag:** 1

**Summary:**

The paper proposed a novel method (Quiet-STaR) that generates rationales at each token to explain the future text, improving their predictions. Quiet-STaR addresses – how to teach LM to produce and use internal thoughts and predict beyond just the next word. They propose a parallel sampling algorithm that operates token by token to make their training procedure scalable. They introduced custom meta-tokens similar to previous works to teach LM when to generate a rationale. Interestingly, they added a residual module that allows their model to determine how much thought needs to be put into the next token prediction. The paper shows that Quiet-STaR outperforms the zero-shot performance.

**Questions To Authors:**

Questions: 1. How did the authors come up with a token “ - - - ” as start and end token embeddings? Did the authors try any other tokens?
2. It is unclear how thought generation improved the model's performance. One simple explanation could be that the model now has more tokens to generate, which gives it more time to calculate the final answer. It might not be that the model learned to reason better.
3. The improvement distribution section is a bit confusing. How did the authors determine which tokens are difficult?

**Reasons To Accept:**

- The paper is well written, but some details are missing. The motivation behind each model component was well explained.

- The idea of teaching a model to generate a meta-token and then act in a certain manner (in their case, meta-token is “thought” and then generates thoughts) is interesting and novel. It is a step towards making these large models more self-aware and controllable.

**Reasons To Reject:**

The paper lacks a stronger baseline. Vanilla Mistral-7B is generally not great at reasoning with CoT. An instruction-tuned model might be a better baseline.
The paper lacks an ablation study and quantitative analysis. These would substantially improve the understanding of how each component contributes to the model's performance.

---

> ### Author Rebuttal · Authors · 2024-05-28
>
> Thank you for your excellent questions!
>
> First, to the question about baselines, we need to emphasize that the key goal of this paper is to show that language models can improve their general inter-token reasoning ability. We attempted to analyze / control for this in the experiments visualized in Figure 2. In general, this method is complementary to CoT, since the model can generate an inner thought before each token of a CoT (as shown in Figure 5 and Appendix E). We will update our paper to note that we chose a base language model because it allows us to study this problem directly, without introducing side-effects from (typically unreleased) RLHF and chat-style fine-tuning strategies used to fine-tune these models.
>
> As for why the start and end thought initializations were selected, it was indeed somewhat arbitrary. We wanted a single token to indicate the start and end of an intermediate thought, similar to what might be used in an actual text, ideally without imposing too many additional biases. We’ll add a note about this.
>
> > One simple explanation could be that the model now has more tokens to generate
>
> Seeing as this is the setup of the pause token paper (Goyal et al 2023), which finds no improvements in a stronger “pause-finetuning” setup (they finetune directly on the downstream task, while we don’t), this explanation is less likely -- though, of course, it is an important one to consider. Notably, this concern is a key challenge for reasoning-related work, and is related to the problem of faithfulness, i.e. how do we know that the internal processing meaningfully corresponds to the generated thought tokens? We'd be happy to elaborate on this more in our limitations section in a revised final version.
>
> > How did the authors determine which tokens are difficult?
>
> In the context of discussing Figure 7, “difficult” was used to refer to high-prediction-loss tokens, but this could be more explicit. We’ll add “(high loss)” to make this clearer.

---

> > ### Comment · Reviewer_G2Zv · 2024-06-07
> > **Thank you for your response**
> >
> > I am happy with the following response:
> > 1. "One simple explanation could be that the model now has more tokens to generate."
> > 2. "How did the authors determine which tokens are difficult?"
> >
> > I'm unsure how start and end thought tokens were selected, and the paper needs more ablation.
> >
> > I am increasing the score to 5-->6, assuming the authors will add the above details in the camera-ready version of the paper.

---

### Decision · Program_Chairs · 2024-07-10

**Decision:**

Accept

**Comment:**

This paper proposes a novel method called Quiet-STaR, that trains the LM to automatically produce internal thoughts that benefit the next token predictions on any arbitrary text. The paper is well-written, and the improvements over two tasks (GSM8K and CommonsenseQA) are significant.

Pros:
- All reviewers agree this work is novel and is an important step towards the general direction of LMs "generating internal thought".
- The method introduces innovative designs such as parallel sampling, mixing heads, and non-myopic losses.

Cons:
- Some reviewers mention the experiments are relatively weak as only one base model (Mistral-7B pre-trained) and one baseline method (CoT) was used.

Overall I think this is a good paper and presents an important step towards making LMs think more in a scalable way, thus I'm voting accept.